# Pelvic Floor Dysfunctions and Their Rehabilitation in Multiple Sclerosis

**DOI:** 10.3390/jcm11071941

**Published:** 2022-03-31

**Authors:** Maddalena Sparaco, Simona Bonavita

**Affiliations:** Department of Advanced Medical and Surgical Sciences, University of Campania Luigi Vanvitelli, Piazza Miraglia 2, 80138 Naples, Italy; ms.86@hotmail.it

**Keywords:** multiple sclerosis, pelvic floor, rehabilitation

## Abstract

Urinary, bowel, and sexual dysfunctions are the most frequent and disabling pelvic floor (PF) disorders in patients with multiple sclerosis (MS). PF dysfunction negatively impacts the performance of daily living activities, walking, and the physical dimension of quality of life (QoL) in people with MS. Patient-reported outcomes on sphincteric functioning could be useful to detect PF disorders and their impact on patients’ lives. PF rehabilitation proposed by Kegel is based on a series of regularly repeated exercises for “the functional restoration of the perineal muscles”. Over time, various therapeutic modalities have been added to PF muscles exercises, through the application of physical or instrumental techniques, such as intravaginal neuromuscular electrical stimulation, electromyographic biofeedback, transcutaneous tibial nerve stimulation. PF rehabilitation has been applied in MS treatment, with improvements of lower urinary tract symptoms severity, QoL, level of anxiety and depression, and sexual dysfunction. This review aims to examine the different PF disorders in MS to evaluate the application of PF rehabilitation in MS and to highlight its advantages and limits, suggesting a multidisciplinary management of PF disorders, with a well-deserved space reserved for PF rehabilitation.

## 1. Introduction

Multiple sclerosis (MS) is a multifactorial demyelinating disease characterized by a large spectrum of symptoms and signs, due to the involvement of the central nervous system (CNS) [1]. The sphincter functions are included in the functional systems (FS) of the Expanded Disability Status Scale (EDSS) [2]. Sphincter dysfunctions include both urination and defecation disorders; sexual function can be documented as well, but it does not impact the FS score, because of assessment difficulties by the examining physician [3].

Urinary, bowel, and sexual dysfunctions are included in the pelvic floor (PF) disorders of MS. Urinary dysfunctions in MS include urgency, increased urinary frequency, and urge incontinence (linked to an overactive bladder), urinary retention, voiding dysfunction with post-void residue (linked to obstructive symptoms). Bowel dysfunctions include constipation or fecal incontinence. Sexual disorders include reduced libido, erectile and ejaculatory dysfunctions, decreased vaginal lubrication and clitoral erection.

A North American Research Committee on Multiple Sclerosis (NARCOMS) survey, conducted on 14,268 patients, demonstrated that moderate-to-severe PF symptoms were reported by one-third of people with MS (pwMS) (bladder, 41%; bowel, 30%; sexual, 42%) [4], negatively impacting on the performance of daily living activities, walking, and the physical dimension of quality of life (QoL) of pwMS [5].

Both in the general population and in pwMS, patient-reported outcomes about sphincteric function could be useful to describe the presence and the impact of PF disorders on patients’ lives.

PF rehabilitation may involve several rehabilitation approaches such as muscle floor retraining, biofeedback, and electrical stimulation of the PF and of the functionally associated musculature. In MS, PF rehabilitation-integrated programs have been demonstrated to play a significant role in patients’ management [6]. This review aims to: (i) examine the different PF disorders in MS; (ii) describe the patient-reported outcomes used for PF disorders, and (iii) evaluate the application of PF rehabilitation in MS, highlighting its advantages and limits. 

## 2. Materials and Methods

### Search Strategy and Selection Criteria

A literature search for articles from 1948 to 2021 was conducted in the databases PubMed and Scopus using the following Medical Subject Headings (MeSH), terms, and key words (also in combination): “multiple sclerosis”, “biofeedback”, “pelvic floor” “constipation”, “fecal incontinence”, “anorectal dysfunction”, “urinary incontinence”, “bowel”, “sphincter”, “bladder” “pelvic floor exercises”, “pelvic floor dyssynergia”, “pelvic floor muscles training”, “overactive bladder”, “voiding dysfunction”, “sexual dysfunctions”. 

Inclusion criteria: (a)Population of interest: adults with MS with bladder, bowel, or sexual dysfunctions;(b)Intervention: rehabilitation program, including physical exercise and/or instrumental techniques;(c)Outcomes: symptoms, impact on quality of life, improvement in self-reported questionnaires, pad test, urodynamic evaluation;(d)Design: prospective and retrospective studies, randomized controlled trials;

Exclusion criteria: animal studies, diseases other than MS, symptoms other than urinary, bowel, or sexual dysfunction. 

Any duplicate studies were also excluded.

The relevant articles were identified and located individually in PubMed/MEDLINE to examine citing and cited-by articles. Two authors independently assessed the articles for relevance. The final reference list was generated based on the relevance to the topics covered in this review, including articles on the prevalence and anatomic aspects of urinary, anorectal, and sexual disturbances in MS, studies on the development and validation of patient-reported outcomes used for PF disorders, and articles that specified the application of PF rehabilitation in pwMS (see PRISMA diagram in Figure 1). Seven articles were included after the revision process.

## 3. Pelvic Floor Dysfunctions in MS

### 3.1. Urinary Dysfunctions in MS

Urinary symptoms in MS are linked to hyperreflexia, hypo contractility, and dyssynergia of the detrusor–sphincter [7]. Bladder dysfunctions in MS depend on spinal cord lesions disconnecting the frontal and pontine micturition centers from the sacral center of the spinal cord [8].

Lesions above the pontine micturition center reduce inhibition and, consequently, lead to detrusor overactivity. Cervical and thoracic spinal cord lesions reduce the central inhibition via damage to the sensory afferent pathways and the pyramidal tract, resulting not only in detrusor hyperactivity but also in a dyssynergia between detrusor contraction and sphincter relaxation. Moreover, pyramidal tracts damage leads to spasticity of the striated sphincters [9]. 

As Ghezzi et al. pointed out, pwMS with signs of pyramidal dysfunction, long disease duration, and higher EDSS, even if asymptomatic for sphincter dysfunctions, should be considered at risk for lower urinary tract symptoms (LUTS), therefore needing further evaluation work-up or, at least, deserving a rigorous follow-up [10].

Urgency, urinary frequency, and urge incontinence are symptoms of overactive bladder and are reported in 37 to 99% of pwMS. Urinary retention and voiding dysfunction with post-void residue are obstructive symptoms, afflicting 34% to 79% of pwMS. A mixed urinary disorder, characterized by the coexistence of an overactive bladder and voiding dysfunction, is present in 50–60% of pwMS [11]. 

In MS, the most common urodynamic findings are detrusor overactivity (mean occurrence of 65%, range 34–99%) or underactivity (mean occurrence of 25%, range 0–40%) and poor bladder compliance (2–10%). Detrusor sphincter dyssynergia is observed in 35% of the patients [11].

### 3.2. Anorectal Dysfunctions in MS

Bowel and bladder dysfunctions are often linked; indeed, constipation and encopresis may also contribute to the development of overactive bladder symptoms and recurrent urinary tract infections (UTIs) that could be the consequence of both constipation and fecal incontinence. Indeed, a full rectum may displace the bladder, leading to its incomplete voiding and subsequent stagnation of urine; on the other hand, encopresis may favor urinary tract colonization. Moreover, UTIs could lead to the exacerbation of bladder instability and enuresis [12,13].

Anal manometry is one of the most important tools for the assessment of the anorectal function and allows studying the anomalies of the anal muscle in MS [14], so that, by understanding the physiopathology, appropriate treatments and targeted rehabilitation therapy could be planned.

Marola et al. found that pwMS with constipation have greater sphincter hypotonia at rest and during contraction compared with constipated non-MS controls, and pwMS with fecal incontinence have lower rectal sensitivity than incontinent controls without MS. The authors concluded that the decrease in the difference in resting anal pressure before and after maximum squeeze maneuvers suggests post-contraction sphincter spasticity, indicating impaired PF coordination in pwMS [15].

Moreover, maximal pressure is lower in progressive compared with relapsing–remitting forms of MS [16].

These studies confirmed previous results indicating a correlation between manometric anomalies and pudendal nerve motor latency in pwMS with constipation or fecal incontinence compared to constipated or incontinent non-MS controls; pwMS with fecal incontinence have lower resting anal pressure compared to non-MS controls, and all pwMS (with and without incontinence) have lower maximum squeeze pressure and higher external anal sphincter fiber densities compared to non-MS controls. Pudendal nerve latency is altered in non-MS controls with fecal incontinence but not in pwMS. These results provide indirect evidence that the anorectal disorders in MS are related to lesions in the CNS [17].

Preziosi et al. confirmed the involvement of the CNS in anorectal disorders in MS, since the rectal anomalies were secondary to spinal cord involvement with rectal compliance correlating with disability. The authors suggested that, in patients with neurologic impairment, rectal compliance is a surrogate of the reflex activity of the spinal cord regulating rectal function, a potential predictor of outcome, and a target for treatment [18].

In pwMS, the prevalence of constipation ranges from 17 to 94%, fecal incontinence from 1 to 69%, and a mixed anorectal dysfunction from 6 to 52% [19].

### 3.3. Sexual Dysfunctions in MS

Sexual dysfunctions (SD) in MS recognize multiple causes, i.e.,

-primary causes, related to direct neurological damage due to demyelinating lesions (i.e., impaired genital sensation), decreased sexual desire, and orgasmic dysfunctions;-secondary causes, as a consequence of MS-related physical changes, such as spasticity, pain, fatigue.-tertiary causes, linked to psychosocial and cultural aspects, which interfere with sexual satisfaction, such as mood disorders or impaired partner relationships [20].

SD have a prevalence of 40–80% in women and 50–90% in men with MS. The most frequent and gender-specific symptoms are erectile and ejaculatory dysfunctions for men, decreased vaginal lubrication, disturbed clitoral erection, and painful intercourse for women [20]. The most frequent SD in both genders is reduced libido.

The management of SD in MS is quite difficult; therefore, besides pharmaceutical intervention and psychological support programs, alternative forms of treatment have been suggested. Recently, a review by Bahmani and Motl [21] highlighted the positive effect of physical exercise on SD in people with MS. The authors suggested several possible mechanisms to explain this beneficial effect: for example, regular physical activity decreases depressive symptoms and fatigue severity, is associated with higher self-esteem, lower feelings of pain, and restorative sleep, and thus, it may positively impact the secondary and tertiary component of SD: Furthermore, neurophysiological changes due to exercise training may favor sexual drive and satisfaction.

## 4. Sphincteric Patient-Reported Outcomes

### 4.1. Urinary Dysfunctions

In 1998, during the first International Consultation on Incontinence (ICI), sponsored by the WHO and organized by the International Continence Society and International Consultation on Urological Diseases, the Scientific Committee recognized the need to develop a questionnaire to assess urinary incontinence in clinical practice and research. The first one was the ICI-Questionnaire (ICIQ) Short Form for urinary incontinence. Other ICIQs have been developed or adapted for urinary, vaginal, and bowel symptoms [22].

Apart from the ICIQ, several scales have been developed and validated to assess bladder dysfunctions and their impact on QoL (Table 1):

OverActivity of the Bladder Questionnaire (OAB-q) [23] and its short form (OABq-Short Form [24]) and very short form (OAB-V8 [25]).Actionable Bladder Symptom Screening Tool (ABSST) [26] (specifically designed for pwMS with urinary incontinence, to help identify who may need and benefit from assessment and treatment).Neurogenic Bladder Symptom Score (NBSS) [27] (designed to assess bladder symptoms and consequences among patients with neurogenic bladder due to neurological disease or lesions, such as MS, spinal cord injuries, or spina bifida). The authors validated each domain as an independent subscale, so to use them not only in combination but also separately [28].Qualiveen (focused on four aspects: bother with limitations, frequency of limitations, fears, and feelings) [29] and its short form (Qualiveen-Short Form [30]).International Prostate Symptom Score (IPSS) [31].

### 4.2. Anorectal Dysfunctions

As already described, anal manometry is one of the most important tools for the assessment of anorectal function. No bowel questionnaire has been validated specifically in pwMS [32]; however, the Neurogenic Bowel Dysfunction Score (NBD) has been created for people with neurogenic bowel dysfunctions [33] and is also used for patients with MS [34]; similarly, the Wexner Constipation and the Wexner Incontinence questionnaires [35,36] are used to evaluate bowel problems in MS [32] (Table 1).

### 4.3. Sexual Dysfunctions

Several questionnaires are used to evaluate sexual problems in MS (Table 1):The Sexual Dysfunction Management and Expectations Assessment in Multiple Sclerosis Female (SEA-MS-F) [37] (developed to ascertain women’s expectations concerning the treatment of sexual dysfunction),The Multiple Sclerosis Intimacy and sexuality questionnaire (a 19-item version [38] and a 15-item version [39], to investigate how various MS symptoms interfere with sexual activity or satisfaction),The Female sexual function questionnaire (SFQ-28) [40], organized into seven domains of female sexual function: desire, physical arousal–sensation, physical arousal–lubrication, enjoyment, orgasm, pain, and partner relationship. Scores for Desire, Arousal, Orgasm, Pain, and Enjoyment are subdivided into three categories that include a high probability of sexual dysfunctions, borderline sexual function, and high probability of normal sexual function. Partner domain and total score are not subdivided into categories, but higher scores indicate better relationships and so less sexual dysfunctions.The Female Sexual Function Index (FSFI) [41].

## 5. Pelvic Floor Rehabilitation in MS

In 1948, Kegel proposed a PF muscles training (PFMT) through a series of regularly repeated exercises for “the functional restoration of the perineal muscles” [42].

As Bø described, there are three proposed theories based on PFMT [43]:(a)a behavioral construct, to learn how to consciously pre-contract the PF muscles before and during increases in abdominal pressure to prevent leakage and(b)two constructs based on changing the neuromuscular function and morphology:strength training builds up long-lasting muscle volume and thus provides structural support;abdominal muscle training indirectly strengthens the PF muscles.



Over time, various therapeutic modalities were added to PF muscles exercises through the application of physical or instrumental techniques, such as biofeedback, electrical stimulation, vaginal cones, hypopressive abdominal gymnastics.

Although PF rehabilitation has been applied for MS treatment for many years [44], there is no standardized protocol used for PF dysfunction, in terms of either PFMT duration or treatment approach.

The PFMT duration in each study was different, with most of the studies having a 12-week duration [45,46,47,48], and a minority lasting 6 weeks [49], 9 weeks [50,51], or 6 months [52,53].

Some studies investigated PFMT alone, with or without physiotherapist guidance [46], or PFMT in addition to other devices or methods, such as intravaginal neuromuscular electrical stimulation (NMES), electromyographic (EMG) biofeedback [49,50], transcutaneous tibial nerve stimulation (TTNS) [47,49,54,55].

Although with different study protocols, PFMT presents several advantages: it is associated with improvements in LUTS severity [45,46,47,48], QoL [45,46,52], level of anxiety and depression [52], and sexual dysfunctions [54,55,56].

In particular, women with MS treated with PFMT reported less storage and voiding symptoms than the sham group [45] and a significant reduction in pad weight, frequency of urgency, and urge urinary incontinence episodes, improvement in all domains of the PF muscles assessment, with lower scores on the OAB-V8 and ICIQ-Short Form [47], reduced number of used pads and nocturia events and improvements in muscle power, endurance, resistance, and fast contractions of PF muscles [48], level of anxiety and depression [52], and arousal, lubrication, satisfaction, and total score domains of the FSFI questionnaire [54] compared with baseline levels.

Moreover, some studies demonstrated the positive effect of adding NMES to PFMT in MS. NMSE causes a reflex PF contraction by stimulating pudendal afferents, with the same inhibitory effect on detrusor activity provoked by a voluntary contraction of PF muscles [9].

Lúcio et al. showed that women with MS treated with the combination of PFMT and intravaginal NMES reported a significantly greater improvement of tone, flexibility, ability to relax the PF muscles, and OAB-V8 scores when compared to subjects treated with PFMT with only EMG biofeedback or with PFMT with EMG biofeedback and TTNS [47].

Lúcio’s study confirmed previous results by Mc Clurg et al. about the benefit obtained by the addition of NMES to a program of PFMT and EMG biofeedback; indeed, there was a significant improvement in the pad test [48] and a reduction in the mean number of leaks (reduced by 85% in the group treated with PFMT+ EMG Biofeedback+ NMES, versus 47% in the group treated with PFMT + EMG biofeedback) [51].

Women with MS treated with PFMT and electrotherapy had a greater improvement of overactive bladder symptoms, perineal musculature contraction [52], and QoL [53] compared with patients treated with PFMT without electrotherapy.

During stimulation, 85% of pwMS were symptoms-free but at three months after treatment cessation, only 18% remained such, although the symptoms were not as pronounced as before treatment. The author suggested maintaining the chronic stimulation to retain the improvement [57].

In this perspective, it would be interesting to have data on the benefit of long-term protocols of PFMT.

Moreover, Perez et al. demonstrated leakage reduction, improvement of QoL (measured using the second part of the OABQ-SF scale and the third question of the ICIQ-Short Form scale), urinary incontinence severity (sum of the values obtained for questions 1–3 on the ICIQ-Short Form), and LUTS (first part of the OABQ-SF scale) after 12 weeks of PFMT both in the group with physiotherapist guidance and in the group without it [45]. Comparing PFMT with and without physiotherapist guidance, there was no difference in leakage reduction and treatment adherence, although the guided PFMT group showed a trend towards better treatment compliance [46].

These findings suggest the need for future studies on PFMT online programs, intended for patients who, owing to their disability, may have difficulties in reaching centers to take part in traditional rehabilitation.

In these patients, self-administered measures of disability and PF function could allow detecting disease progression over time, capturing treatment effects, and could help select patients to include in PF rehabilitation programs [58].

The literature about conservative interventions on anorectal dysfunctions in MS is scanty, and most of the studies mainly focused on lifestyle advice and bowel biofeedback techniques.

It is not yet clear whether instrumented biofeedback training is useful or whether, as suggested by Norton et al. [59] for the general population, lifestyle advice and patient education alone would be sufficient. In a randomized controlled trial, Norton et al. [59] tested four approaches in patients with fecal incontinence: (1) medical advice; (2) advice and sphincter exercises; (3) hospital-based computer-assisted sphincter pressure biofeedback; (4) hospital biofeedback plus home biofeedback. The authors found that neither PF exercises nor biofeedback were superior to advice and education.

However, as Schott et al. [60] pointed out, this trial, having a sample size too small, did “not have the power to conclude” on the effects of these treatments and might be misleading.

Although there are no data to conclude whether advice on the correct lifestyle and patient education has a superior effect on bowel dysfunctions than PFMT and biofeedback in MS, some lifestyle recommendations may be advantageous, such as the adoption of a high-fiber diet, high fluid intake, and regular bowel movement [13]. This aspect is very important, considering the role of the gut microbiome and gut dysbiosis in MS; indeed, besides determining diarrhea or constipation, dysbiosis may compromise the integrity of the so-called “gut barrier”, leading to the leaky gut syndrome that can provoke systemic and neuroinflammation that, in turn, by affecting efferent cholinergic transmission, could result in intestinal inflammation [61]. Despite further studies being necessary to clarify the “direction of the Gut–Brain Axis”, it is known that in MS (and in a murine model of the disease, i.e., experimental autoimmune encephalomyelitis) gastrointestinal symptoms and/or an altered gut microbiota have been reported together with increased intestinal permeability. Compared to healthy controls, MS patients have a decrease in the proportion of *Faecalibacterium*, *Eubacterium rectale*, *Corynebacterium*, *Fusobacteria* and an increase of *Escherichia*, *Shigella*, *Clostridium*, Firmicutes [62,63]. Decreased numbers of *Faecalibacterium* spp. and lower levels of their metabolite butyrate lead to a decrease of Treg cells, antigen-presenting cells, and pro-inflammatory cytokines [64]. Several authors reported a decrease of such *Bacteroides* spp. as *Bacteroides stercoris* and *Bacteroides coprocola* in the gut microbiota of MS patients and a negative correlation between the number of *Prevotella copri* and the risk of MS development [65]. All these data support a possible role of the microbiota in the pathogenesis and/or progression of the disease.

However, studies support the bowel biofeedback treatment to improve anorectal dysfunctions in pwMS [17,32,66].

Preziosi et al. [32] aimed to identify the effect of biofeedback on bowel symptoms, mood, and anorectal physiology in pwMS. The authors reported a significant improvement of constipation and fecal incontinence (*p* < 0.001) after bowel biofeedback therapy, with a response rate of 46%. Furthermore, the higher was the initial bowel symptoms score, the greater was the improvement. The greater improvement of endurance squeeze pressure was obtained in responders compared to no responders (*p* = 0.008). The authors also demonstrated a significant improvement of depression (using the Hospital Anxiety and Depression Scale, *p* = 0.015), although a specific QoL test was not conducted. However, the absence of a control group induces to be cautious in the interpretation of these results.

Munteis et al. [17] investigated 18 pwMS with constipation and fecal incontinence performing manometric biofeedback: 44.4% of them reported a significant improvement of anorectal dysfunction (6 complete, 2 partial), presenting milder manometric abnormalities (though not significantly different) than patients without improvement.

When tailoring a rehabilitation program, it is important to consider factors predictive of symptom improvements such as mild to moderate disability, quiescent and non-relapsing disease, and absence of progression of MS over the year before the biofeedback treatment [66].

## 6. Limitations

PwMS with PF disorders may present [67]

a high coexistence of bowel and bladder dysfunctions;a coexistence of mixed sphincter dysfunctions (retention plus urgency or incontinence);absence of correlation between the pattern of bowel symptoms and urinary disturbance.

Although PFMT showed several advantages in comparison to no treatment or inactive control treatments, we identified several caveats:absence of a consensus on the protocol to use to manage PF dysfunction,absence of a uniform approach to PF exercises,different devices or methods used in addition to PF training,different evaluation of PF dyssynergia in pwMS (clinical parameters, patient-reported outcomes, EMG activity, manometry),a shortage of studies on anorectal dysfunctions treatment,absence of data on the long-term benefit of PFMT.

Therefore, further studies are necessary to (1) investigate the effect of a correct educational policy on bowel and urinary dysfunction, (2) optimize PF exercises, (3) standardize PFMT protocols and approaches with devices or methods used in addition to PF exercises, (4) obtain data on the long-term benefit of PFMT.

In our opinion, to better understand the mechanisms underlying improvements (or failures) after PF rehabilitation, it would be interesting to evaluate if a functional reorganization occurs in the brain neuronal networks after PFMT, as it happens after motor [68] and/or cognitive rehabilitation [69,70].

## 7. Discussion and Conclusions

Although studies on the management of PF dysfunction did not use standardized protocols and enrolled heterogeneous patients, they were mostly based on a “conservative” approach, focusing on the pelvic problem and planning a correct education program and an appropriate training based on PF exercises and/or biofeedback, NMES, TTNS.

The conservative approach alone has resulted in a significant relief of symptoms related to sphincter dysfunction, with an important impact on QoL.

As a Cochrane systematic review showed, there is moderate-quality evidence that inpatient or outpatient multidisciplinary rehabilitation programs improve bladder dysfunctions in pwMS [71]; therefore, the best approach for pwMS might be a multidisciplinary rehabilitation program that considers the large spectrum of symptoms and signs that characterize MS, including PF dysfunctions, with PFMT as a part of the complete program.

As previously reported by a Cochrane systematic review on PFMT in women with PF dysfunctions in the general population, PFMT could be similarly included in first-line conservative management programs for women with MS either to gain clinical improvement or for the promising cost–effectiveness ratio [72].

A UK consensus on the management of the bladder in MS establishes that there is level II b evidence about the effects of PFMT in MS, suggesting that PF excises “may be effective and there is certainly no evidence that these can be harmful”. Moreover, a recommendation of the UK consensus established that “PF exercises should be offered to patients with mild disability from MS”, suggesting an assessment of PF contractions before initiating the treatment [9].

Therefore, we suggest a step-by-step diagnostic evaluation and therapeutic program.

The first diagnostic step is screening for PF disorders, also in a- or paucisymptomatic pwMS, using a self-administered scale (PROs) considering bladder, bowel, and sexual functions.The second diagnostic step is a specific assessment, based on the results of PROs, considering urine testing (if pwMS presented UTI symptoms), abdominal ultrasound, and, in the second line, urodynamics and manometric exams. We highlight, at this step, the importance of a correct assessment of PF contractions with a digital technique, based on the PERFECT scheme: P meaning power (or pressure), E, endurance, R, repetitions, F, fast contractions, and finally, ECT, every contraction timed. The use of a perineometer could help in this assessment [73].(1)The first therapeutic step is based on a conservative approach: adequate diet and lifestyle and/or a pharmacologic approach (for example, with alpha-blockers, antimuscarinic anti-diarrheic agents, prokinetics), psychological assessment, and physiotherapeutic evaluation to perform a correct PFMT (in particular, in pwMS with mild disability), with or without biofeedback, NMES, TTNS.(2)The second therapeutic step is based on a progressively more invasive approach: from intermittent self-catheterization and/or anal irrigation to detrusor injection of botulinum toxin A or sacral neuromodulation.

Although in the last years there has been increasing attention on PF disorders in MS, their management remains an open issue. The NARCOMS study evaluated pwMS satisfaction with the current evaluation and treatment of PF disorders: most respondents were moderately to very satisfied with the management of their bladder and bowel disorders, but significantly less satisfied with their SD care [4].

Therefore, to make pwMS more satisfied with the care of all the aspects of PF disorders, we suggest a holistic view of the PF disorders and a consequent multidisciplinary management based on neurologic, gynecologic, physiotherapeutic, urologic, gastroenterological, nutritional, and psychological assessments, with the deserved attention payed to PFMT.

## Figures and Tables

**Figure 1 jcm-11-01941-f001:**
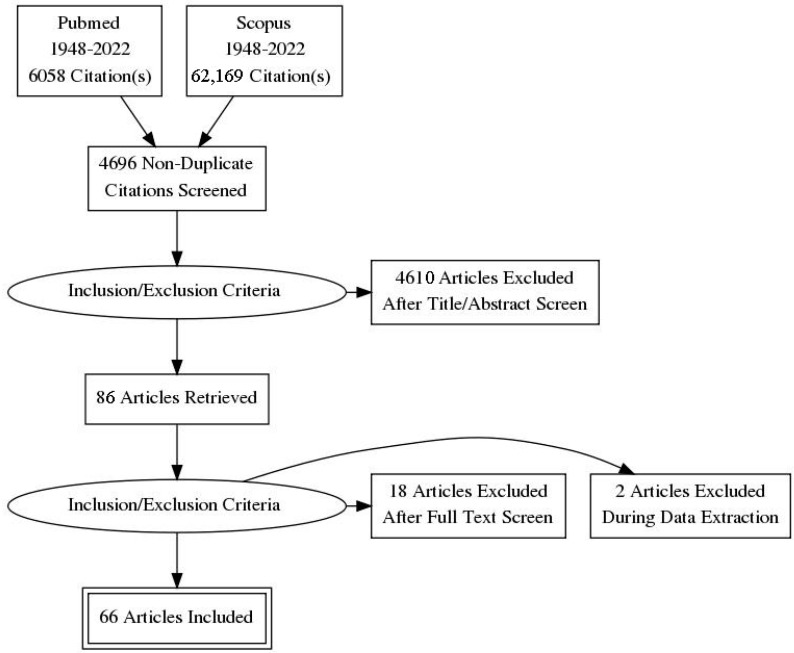
Flowchart of the literature selection process.

**Table 1 jcm-11-01941-t001:** Patient-reported outcome questionnaires for sphincteric and sexual dysfunction.

Questionnaire	Items	Score	Signification	Notes
URINARY DYSFUNCTIONS				
ICI-Q UI SF	4	0 to 21	higher scores: greater severity of symptoms	
OAB-q	33	0 to 100	higher scores: greater severity of symptoms and lower QoL	Two short forms: OABq-SF and OAB-V8
ABSST	8	≥3 need for further urogynecological evaluation and treatment		
NBSS	25	0 to 74	higher scores: greater severity of symptoms	Three domains: Incontinence, Storage and Voiding, and Consequences (used not only in combination but also separately)
Qualiveen	30	for each domain 0: no effect of urinary problems on QoL 4: high impact on QoL	higher scores: higher QoL	Short form: 8 items
IPSS	7	0–7—mild 8–19—moderate 20–35—severe	measure of frequency and severity of symptoms	An additional item measures the impact on QoL
ANORECTAL DYSFUNCTIONS				
NBD	10	0 to 47	higher score: higher severity of dysfunction	
Wexner incontinence score	5	0 to 20	0: absence of symptoms, 20: highest severity of symptoms	
Wexner constipation score	8	0 to 30	0: absence of symptoms, 30: highest severity of symptoms	
SEXUAL DYSFUNCTIONS				
SEA-MS-F	8	0 to 32	organized into 3 parts: general expectations (sexuality); specific expectations (sexual symptoms); ultimate goals for treatment of sexual dysfunction	
MSISQ-19	19	19 to 95	higher scores: greater impact of MS symptoms on sexual life	Specific subscale (used also separately) for the primary, secondary, and tertiary aspects of sexual dysfunctions in MS
MSISQ-15	15	15 to 75	higher scores: greater impact of MS symptoms on sexual life	Specific subscale (used also separately) for the primary, secondary, and tertiary aspects of sexual dysfunctions in MS
SFQ-28	28	each of the 7 domains has a different score range, indicating (from the lower to the higher scores) high probability of sexual dysfunction, borderline status, and normal sexual function		
FSFI	19	2 to 36	Higher scores indicate better sexual functioning	Six domains: sexual desire, sexual arousal, lubrication, orgasm, satisfaction, and pain

Legend to Table 1: International Consultation on Incontinence-Questionnaire for Urinary Incontinence Short Form (ICI-Q UI SF), OverActivity of the Bladder Questionnaire (OAB-q), Actionable Bladder Symptom Screening Tool (ABSST), Neurogenic Bladder Symptom Score (NBSS), International Prostate Symptom Score (IPSS), Neurogenic Bowel Dysfunction Score (NBD), Sexual Dysfunction Management and Expectations Assessment in Multiple Sclerosis Female (SEA-MS-F), Multiple sclerosis intimacy and sexuality questionnaire 19-item version (MSISQ-19) and a 15-item version (MSISQ-15), Female Sexual function Questionnaire (SFQ-28), Female Sexual Function Index (FSFI).

## Data Availability

Not applicable.

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
