# Peer review of "Pelvic Floor Dysfunctions and Their Rehabilitation in Multiple Sclerosis"

_jcm, 2022, doi:10.3390/jcm11071941_

Round 1

Reviewer 1 Report

In this review article, the authors address a very important topic in multiple sclerosis (MS) clinical management in a comprehensive fashion – the evaluation and management of pelvic floor dysfunction. The review is well-organized and explains clinical concepts in an approach that is easy to understand. However, I have a few major comments that I would like to raise. The article appears to be missing important information about the role of the gut microbiome and gut dysbiosis as another possible etiologic factor for bowel dysfunction and symptomatology in MS. I recommend inclusion of this topic in the results and discussion. Several paragraphs in the article also appear to be redundant and just relist information already present in Table 1 and should be removed or rephrased.

Please see more specific comments below:

Abstract:

This can be reworded or expanded further if word count allows to mention the role and benefit of PFMT, a topic of major focus in the overall review article which is not currently mentioned here.

Introduction:

  • Line 20, “Being sphincter dysfunctions very common and disabling in MS, …”

This sentence is not clearly or correctly worded and should be revised.

  • Line 26, “conducted on 14.268”

Authors should use a comma as a delimiter instead of a decimal point.

Materials and Methods:

  • Line 46, “articles on the epidemic and anatomic aspects of urinary, anorectal, and sexual disturbances in MS”

Epidemic is a misnomer, please correct.

Urinary dysfunctions in MS:

- Line 61, “The bladder dysfunction in MS depends on the spinal cord lesions disconnecting the frontal and pontine micturition centers from the sacral center of the spinal cord.”

This line and the following paragraph should be listed at the beginning of this section to explain the pathophysiology of urinary symptoms in MS first before describing their prevalence and urodynamic findings. Same for the “Anorectal dysfunctions in MS” section.

Anorectal dysfunctions in MS:

  • Line 98, “no-MS controls”

Please revise to “non-MS controls” or “controls without MS”. Please specify if healthy controls or patients with other pelvic floor disorders for each of these studies.

Sexual dysfunctions in MS:

  • Line 112, “disagreeable gender-specific symptoms”

What does “disagreeable” mean? This appears to be redundant with gender-specific and does not add to the sentence.

Urinary dysfunctions

  • Lines 143 - 175, “Apart from ICIQ, several scales have been developed and validated to assess bladder dysfunctions and their impact on QoL…”

All of the information mentioned in this and subsequent paragraphs in this section are redundant and just mention information already listed in Table 1. Please rephrase or take out entirely. Same applies to the “Sexual dysfunctions” section lines 197-220.

Other:

There are several other grammatical and typographical errors throughout the manuscript not mentioned above. The manuscript would significantly benefit from English proofreading.

Author Response

In this review article, the authors address a very important topic in multiple sclerosis (MS) clinical management in a comprehensive fashion – the evaluation and management of pelvic floor dysfunction. The review is well-organized and explains clinical concepts in an approach that is easy to understand.

However, I have a few major comments that I would like to raise.

The article appears to be missing important information about the role of the gut microbiome and gut dysbiosis as another possible etiologic factor for bowel dysfunction and symptomatology in MS. I recommend inclusion of this topic in the results and discussion.

We thank the reviewer for this suggestion. We added a paragraph dealing with the role of dysbiosis in MS and on bowel disturbances. (Page 11, lines 374-392)

Several paragraphs in the article also appear to be redundant and just relist information already present in Table 1 and should be removed or rephrased.

According to the reviewer request, we rephrased the paragraphs linked to Table 1.

Please see more specific comments below:

Abstract:

This can be reworded or expanded further if word count allows to mention the role and benefit of PFMT, a topic of major focus in the overall review article which is not currently mentioned here.

We thank the reviewer for this suggestion. We expanded the abstract accordingly (Page 1, lines 11-17; 20-22)

Introduction:

  • Line 20, “Being sphincter dysfunctions very common and disabling in MS, …”

This sentence is not clearly or correctly worded and should be revised.

We agree with the reviewer and deleted it.

  • Line 26, “conducted on 14.268”

Authors should use a comma as a delimiter instead of a decimal point.

We corrected it.

Materials and Methods:

  • Line 46, “articles on the epidemic and anatomic aspects of urinary, anorectal, and sexual disturbances in MS”

Epidemic is a misnomer, please correct.

We corrected it.

Urinary dysfunctions in MS:

- Line 61, “The bladder dysfunction in MS depends on the spinal cord lesions disconnecting the frontal and pontine micturition centers from the sacral center of the spinal cord.”

This line and the following paragraph should be listed at the beginning of this section to explain the pathophysiology of urinary symptoms in MS first before describing their prevalence and urodynamic findings. Same for the “Anorectal dysfunctions in MS” section.

We thank the reviewer for the suggestion; as suggested, we moved the paragraph to explain the pathophysiology of urinary symptoms and anorectal dysfunctions in MS at the beginning of each section.

Anorectal dysfunctions in MS:

  • Line 98, “no-MS controls”

Please revise to “non-MS controls” or “controls without MS”. Please specify if healthy controls or patients with other pelvic floor disorders for each of these studies.

We apologize for the oversight, we corrected it.

Sexual dysfunctions in MS:

  • Line 112, “disagreeable gender-specific symptoms”

What does “disagreeable” mean? This appears to be redundant with gender-specific and does not add to the sentence.

We deleted this word.

Urinary dysfunctions

  • Lines 143 - 175, “Apart from ICIQ, several scales have been developed and validated to assess bladder dysfunctions and their impact on QoL…”

All of the information mentioned in this and subsequent paragraphs in this section are redundant and just mention information already listed in Table 1. Please rephrase or take out entirely. Same applies to the “Sexual dysfunctions” section lines 197-220.

We thank the reviewer for the suggestion. We deleted the information included in Table 1 and kept the information that were not reported in table 1.

Other:

There are several other grammatical and typographical errors throughout the manuscript not mentioned above. The manuscript would significantly benefit from English proofreading.

Thank you for the comment; we edited the manuscript

Reviewer 2 Report

The manuscript by Sparaco et al. is a valuable study on the important problem of rehabilitation of pelvic floor dysfunction in patients with multiple sclerosis. Congratulations to the Authors for undertaking research on this subject. Below are some comments on the manuscript that I hope will help to improve it.

  1. I think that the introduction can be slightly expanded to introduce well into the discussed issue. Currently, it is very short and general
  2. also the abstract does not fully reflect the whole manuscript, please complete the it.
  3. please clearly indicate the inclusion and exclusion criteria. It would also be appropriate to include a PRISMA diagram, which will systematize the entire scheme.
  4. did the authors assess the quality of qualified works? Please indicate and do not treat equally articles of different scientific levels.
  5. There is no reference to the literature in some parts of the manuscript (e.g. lines 52-56 - on what basis this paragraph was derived).

Author Response

The manuscript by Sparaco et al. is a valuable study on the important problem of rehabilitation of pelvic floor dysfunction in patients with multiple sclerosis. Congratulations to the Authors for undertaking research on this subject.

We thank the reviewer for the comment. We modified as suggested and the revisions are tracked in revised manuscript.

Below are some comments on the manuscript that I hope will help to improve it.

  • I think that the introduction can be slightly expanded to introduce well into the discussed issue. Currently, it is very short and general

We thank the reviewer for the suggestion. As requested, we have expanded the introduction. (Page 1, lines 34-40; 46-51)

  • also, the abstract does not fully reflect the whole manuscript, please complete it.

As also requested by Reviewer #1, we expanded this part (Page 1, lines 11-17; 20-22)

  • please clearly indicate the inclusion and exclusion criteria. It would also be appropriate to include a PRISMA diagram, which will systematize the entire scheme.

We thank the reviewer for the suggestion. We now inserted the inclusion/exclusion criteria in the methods section (Page 2, lines 62-74) and added a PRISMA diagram

  • did the authors assess the quality of qualified works? Please indicate and do not treat equally articles of different scientific levels.

We thank the reviewer for the comment. Since this review is not a systematic review of the literature rather a narrative one, the scientific quality of the included studies was not formally assessed and documented; however, we reported articles published on peer reviewed journals considering authors’ reputation, accuracy of methods to test the initial hypothesis, key results, limitations, quality, and interpretation of the results obtained, and impact of the conclusions. After this process the studies with the best contributions have been synthetized. The selected articles indeed, have a mean impact factor of 6.5 that could be considered adequate, especially in the rehabilitation field.

  • There is no reference to the literature in some parts of the manuscript (e.g. lines 52-56 - on what basis this paragraph was derived).

We apologize for the oversight; we have now added the missing reference

Round 2

Reviewer 1 Report

I would like to thank the authors for the manuscript revisions which have addressed the majority of my comments and concerns. A few minor points remain:

Abstract: Kegel is misspelled.

Figure 1: please specify the reason(s) for exclusion of articles.

Sexual dysfunctions in MS (Line 156): This paragraph seems to copy/paste what was previously mentioned in the introduction. Please remove or rephrase to avoid redundancy.

Line 321: please change "cytokine" to "cytokines"

Author Response

We Thank the reviewer for the careful revisions.

We now addressed his comments

Abstract: Kegel is misspelled. We corrected the misspell

Figure 1: please specify the reason(s) for exclusion of articles. The articles have been excluded because they did not satisfy the inclusion/exclusion criteria; we believe that this can be guessed by following the arrows in the graph; therefore we have not added this info in the figure to avoid being repetitive 

Sexual dysfunctions in MS (Line 156): This paragraph seems to copy/paste what was previously mentioned in the introduction. Please remove or rephrase to avoid redundancy. We thank the remark for the comment; we simplified the paragraph in the introduction and kept as it is that one in the section "Sexual dysfunctions in MS" .

Line 321: please change "cytokine" to "cytokines". We corrected it
